# Intensive Care Weaning (iCareWean) protocol on weaning from mechanical ventilation: a single-blinded multicentre randomised control trial comparing an open-loop decision support system and routine care, in the general intensive care unit

M P Vizcaychipi [1,2,3] Laura Martins,[4] James R White,[5] Dan Stleper Karbing,[3] Amandeep Gupta,[6] Suveer Singh,[2] Leyla Osman,[5] Jeronimo Moreno-Cuesta,[7] Steve Rees[3]

For numbered affiliations see end of article.

**Correspondence to**
Dr M P Vizcaychipi;
m.vizcaychipi@imperial.ac.uk

## ABSTRACT

**Introduction** Automated systems for ventilator management to date have been either fully heuristic rule-based systems or based on a combination of simple physiological models and rules. These have been shown to reduce the duration of mechanical ventilation in simple to wean patients. At present, there are no published studies that evaluate the effect of systems that use detailed physiological descriptions of the individual patient. The BEACON Caresystem is a model-based decision support system that uses mathematical models of patients' physiology in combination with models of clinical preferences to provide advice on appropriate ventilator settings. An individual physiological description may be particularly advantageous in selecting the appropriate therapy for a complex, heterogeneous, intensive care unit (ICU) patient population.

**Methods and analysis** Intenive Care weaning (iCareWean) is a single-blinded, multicentre, prospective randomised control trial evaluating management of mechanical ventilation as directed by the BEACON Caresystem compared with that of current care, in the general intensive care setting. The trial will enrol 274 participants across multiple London National Health Service ICUs. The trial will use a primary outcome of duration of mechanical ventilation until successful extubation.

**Ethics and dissemination** Safety oversight will be under the direction of an independent committee of the study sponsor. Study approval was obtained from the regional ethics committee of the Health Research Authority (HRA), (Research Ethic Committee (REC) reference: 17/LO/0887. Integrated Research Application System (IRAS) reference: 226610. Results will be disseminated through international critical care conference/symposium and publication in peer-reviewed journal.

**Trial registration number** ClinicalTrials.gov under NCT03249623. This research is registered with the National Institute for Health Research under CPMS ID: 34831.

## Strengths and limitations of this study

► Single-blinded multicentre randomised control trial evaluating management of mechanical ventilation as directed by the BEACON Caresystem compared with that of standard care in the general intensive care setting.

► The BEACON Caresystem is a model-based decision support system that uses mathematical models of patients' physiology in combination with models of clinical preferences to provide advice on appropriate ventilator settings.

► Sample size was determined by a power analysis of the duration of mechanical ventilation.

► Patients and the public were not involved in the design of this study.

► The study does not assess successful rehabilitation of the patient subsequent to their intensive care unit stay.

## INTRODUCTION

Optimising mechanical ventilator management has been the focus of much research, seeking to decrease ventilator-associated complications such as ventilator-associated lung injury, ventilator-associated pneumonia, respiratory muscle atrophy and patient discomfort.[1 2] Increased duration of mechanical ventilation and prolonged weaning increases the number of these complications with a vicious cycle that leads to increased morbidity and mortality and associated economic costs.[2] Protocols and computerised tools aiming to reduce

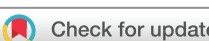

length of mechanical ventilation might therefore be beneficial.

Weaning protocols and nurse-lead ventilator management is routinely used in the intensive care setting.[2][3] Such approaches have been shown to reduce the duration of mechanical ventilation.[3] However, standardising this process is important, and the quality and quantity of nursing care are a critical factor in reducing duration of mechanical ventilation.[4]

Standardisation of care through automation or bedside decision support continues to be driven by novel technological innovation, some of which is already established in clinical practice.[3][5] However, to date, much of this has been predicated on clinical guidelines rather than using detailed physiological descriptions of the individual patient.[6–8] A Cochrane review of weaning trials with current systems concluded that their use may reduce duration of weaning, but highlighted that the population was predominantly composed of those who are 'simple to wean'.[5] The patient populations of these studies were characterised as without additional lung pathology and were predominantly ventilated for periods of less than 48 hours.[5] Consequently, it remains to be investigated whether decision support systems can reduce duration of mechanical ventilation in patients with more than a few hours of 'routine' postoperative intubation. The BEACON Caresystem is a model-based decision support system that uses mathematical models of patients' physiology in combination with models of clinical preferences to provide advice on appropriate ventilator settings.[9] An individual physiological description may be particularly advantageous in selecting the appropriate therapy for a complex, heterogeneous, intensive care unit (ICU) patient population. The BEACON Caresystem is the commercial version of the INVENT system, which has been retrospectively evaluated in postoperative cardiac patients and patients with severe lung disease, and prospectively evaluated in providing advice on correct levels of inspired oxygen.[10–13] To date, the commercial system has been evaluated over short durations of time from 4 to 8 hours to establish whether physiological measurements are improved.[14][15] No previous studies evaluating outcome of weaning from mechanical ventilation have been performed.

Here, we describe a single-blinded multicentre randomised control trial evaluating management of mechanical ventilation as directed by the BEACON Caresystem compared with that of current care in the general intensive care setting.

## METHODS AND ANALYSIS
### Administrative structure
The study is under the auspices of the eHRA, REC reference: 17/LO/0887, IRAS reference: 226610, and with the National Institute for Health Research under

CPMS ID: 34 831. The research and development department of the Chelsea and Westminster National Health Service (NHS) Foundation Trust is sponsor of the study and responsible for study conduct, safety monitoring, personnel training and randomisation. Data collection occurs at participating centres, with data analysis performed in collaboration with the respiratory and critical care group, Aalborg University and independent statisticians of the study sponsor. All data transfer and storage will be performed in accordance with General Data Protection Regulation (GDPR) regulations and the study will monitor and review interim analysis and made recommendations to the steering committee.

### Study design
To test the hypothesis that following advice from the BEACON Caresystem will reduce time to successful extubation, the Intenive Care weaning (iCareWean) group has designed a single-blinded multicentre randomised control trial evaluating management of mechanical ventilation as directed by the BEACON Caresystem compared with that of current care in the general intensive care setting. The Consolidated Standards of Reporting Trials (CONSORT) flow diagram of the study is illustrated in figure 1.

### Description of the BEACON Caresystem
At the core, functionality of the BEACON Caresystem is a set of physiological models including models of pulmonary gas exchange, acid–base chemistry, lung mechanics and respiratory drive.[9] The BEACON Caresystem tunes these models to the individual patient so that they accurately describe the current physiological measurements. Once the models are tuned, they are then used by the system to simulate the effects of a change in ventilator settings. The results of these simulations are then used to calculate the clinical benefit of changing the ventilator settings by balancing the competing goals of mechanical ventilation using mathematical models of clinical preference.[9][10] Appropriate ventilator settings imply a balance between quantified preferences of several competing clinical goals. For oxygenation, the preferred values of inspired oxygen fraction are weighed against the need to prevent hypoxaemia. For settings of ventilation volume, pressures or frequency, the risk of lung trauma is weighed against the need to prevent acidosis. For settings of support in spontaneously breathing patients, the risk of respiratory muscle atrophy on over support is weighed against the excessive work of breathing and potential of respiratory muscle fatigue on under support. Quantification of these risks is used to calculate a total score for any possible ventilator strategy. The system then calculates advice so as to change ventilator settings to improve this score.

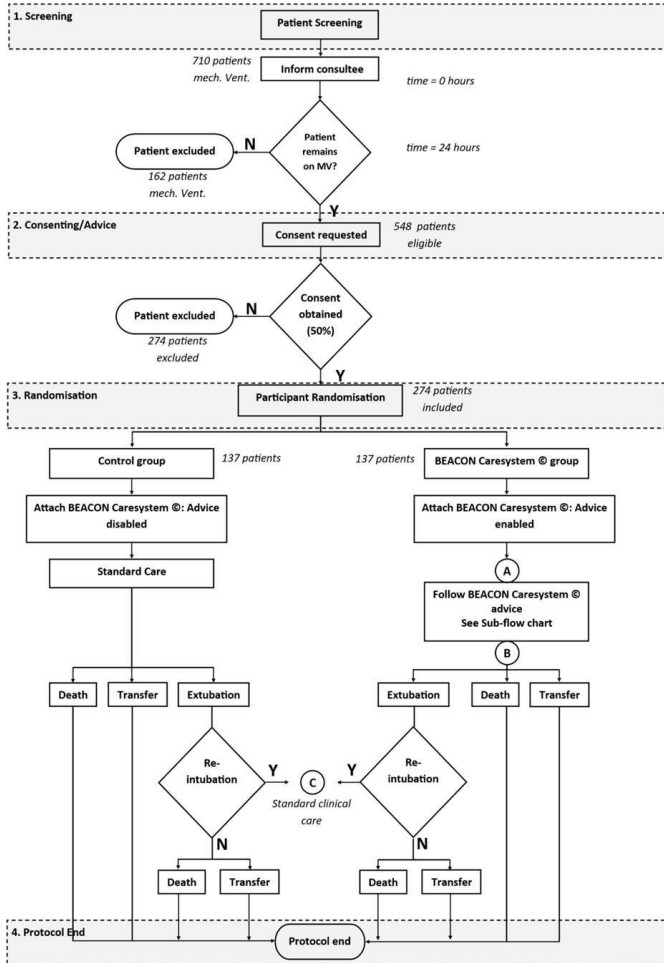

**Figure 1** Consolidated Standards of Reporting Trials flow diagram for protocol. MV, mechanical ventilation.

The BEACON Caresystem functions as an 'open-loop' system. This means that the advice provided by the system is presented to the clinician. The ventilator settings are then changed by the clinician, and the patient's response to these changes is automatically registered and used by the BEACON Caresystem to retune the physiological models and repeat the process of generating new advice.

The BEACON Caresystem provides advice on ventilator settings including the following. Values for maximal advice steps are given in table 1.

► Fraction of inspired oxygen (FiO₂).

► Tidal volume (Vt).
► Respiratory frequency (f).
► Positive end expiratory pressure (PEEP).
► Pressure above PEEP.

## Patient screening

All patients will be screened for eligibility on admission to the ICU by the research team, including research nurses and clinical research fellows. Advice will be obtained from relatives acting as a personal consultee, or consultant acting as nominated professional consultee. Retrospective consent is then obtained from the patient on regaining capacity. The study will run from September 2017 to November 2020.

## Eligibility criteria
### Inclusion criteria

1. At least 24 hours of invasive mechanical ventilation.
2. Age ≥18 years.
3. Patient is mechanically ventilated in a ventilator mode, and by a ventilator, supported by the BEACON Caresystem.
4. Haemodynamically stable, where instability is defined by the presence of two or more of the following criteria: acidosis pH <7.2, poor urine output <0.5 mL/kg/h, use of vasopressors, for example, noradrenaline >25 µg/min.

### Exclusion criteria

1. The absence of an arterial catheter for blood sampling.
2. Medical history of home mechanical ventilation which may lead to prolonged stay in the ICU, including long-term oxygen therapy and non-invasive ventilation not associated with sleep apnoea.
3. Clinical conditions requiring treatment with extracorporeal membrane oxygenation (ECMO), that is, an inspired oxygen of 100% for more than 24 hours.
4. Head trauma or other conditions where intracranial pressure may be elevated and tight regulation of arterial $CO_2$ level is paramount.
5. Primary (non-overdose related) neurological patients (Glasgow Coma Score 24).
6. Severe heart failure classified by grade 4 American Heart Association guidelines.
7. End-stage liver disease.

| Table 1 | Maximal steps in mechanical ventilation advice provided by BEACON Caresystem | |
|---|---|---|
| **Maximal allowed step in ventilator settings** | **Stepping up** | **Stepping down** |
| FIO₂ | 10 % | 5% or 2% when SpO₂ ≤92 % |
| Vt (volume modes) | 50 mL | 50 mL |
| Respiratory frequency (f) | 5 breaths/min or 3 breaths/min when f≥18 | 5 breaths/min |
| Pressure above PEEP (pressure modes) | 2 cm H₂O | 2 cm H₂O |
| PEEP | 3 cm H₂O if PEEP <10 cm H₂O, otherwise 2 cm H₂O | 2 cm H₂O |

f, respiratory frequency; FiO2, fraction of inspired oxygen; PEEP, positive end expiratory pressure; Vt, tidal volume.

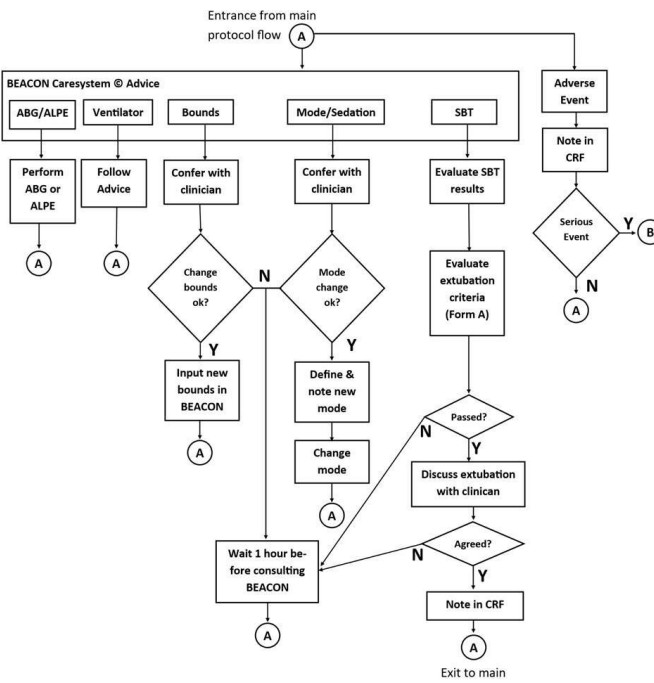

**Figure 2** Flow chart of BEACON Caresystem intervention arm. ABG, arterial blood gas; ALPE, automatic lung parameter estimator; CRF, clinical research form; SBT, spontaneous breathing trial.

8. Multiple ICU admissions within the medical stratification group, that is, more than one admission.
9. Corrective orthognathic surgery.
10. ICU admission following oesophageal surgery.
11. Patients with morbid obesity defined as either body mass index (BMI)≥45; or BMI≥35 with an Acute Physiology and Chronic Health Evaluation II (APACHEII) score on admission >24.
12. Pregnancy.
13. Mechanical ventilation initiated for more than 24 hours in other centres.

### Randomisation
Figure 1 illustrates the CONSORT flow diagram for the study. On recruitment to the trial, participants are randomised to either to the intervention group where advice is provided by the BEACON Caresystem, or to the control arm where standard care is provided. Block randomisation is applied with each block stratified to ensure equal distribution between control and intervention arm for the following patient groups, reducing the risk of potential bias by over-representation of these groups in the control or intervention arms:

► Surgical or medical ICU admission.
► Presence of absence of chronic obstructive pulmonary disease.
► Primary thoracic or non-thoracic medical diagnosis.

### Study procedure
In the intervention arm of the trial, the BEACON Caresystem is activated by the research team and subsequent ventilator management determined by system advice, applied to the ventilator by the clinical team, as illustrated in figure 2 and described below. In the control arm, the BEACON Caresystem is activated in a mode in which system advice is disabled, but data are still collected by the BEACON Caresystem. The ventilator management in this arm is determined by standard care.

### Intervention arm
Figure 2 illustrates the protocol flow in the intervention arm. At any time during the protocol, adverse events are addressed immediately, with a serious event resulting in exclusion from the study (illustrated as B in figure 2). Minor events are documented and the patient returned to the main protocol flow (illustrated at point A, at the top of figure 2). During intervention, the BEACON Caresystem can provide a number of different forms of advice to the user which require action. These are illustrated at the top of figure 2, and are described in turn below.

### Arterial blood gas (ABG)/automatic lung parameter estimator (ALPE)
In situations where mathematical model simulations cannot accurately describe continuous measurements, that is, $SpO_2$, fraction of end-tidal $CO_2$ ($FetCO_2$), or in support ventilator modes respiratory frequency or tidal volume, then retuning of the models is required. In this situation, the BEACON Caresystem requests either an ABG, or a short procedure varying $FiO_2$ to assess pulmonary gas exchange (ALPE), as described and evaluated previously,[16–20] and briefly described in the online supplementary material.

### Ventilator
Advice on changing ventilator settings. New advice is generated only when the system regards current settings as not appropriate. Duration between advice therefore varies, however, for PEEP and changes in volume or pressure 10–20 min is the minimum period between advice.

### Bounds
The BEACON Caresystem allows for bounds to be set for the limits of advice on ventilator settings. If advice is available which is outside these bounds, the protocol dictates that the bedside nurse consults with the clinician on duty.

### Mode/sedation
If the BEACON Caresystem detects substantial spontaneous breathing during control ventilator modes, advice is provided to consider changing to support mode. Additionally, if pH falls below a defined value in support mode, advice is provided to consider changing to control mode or modifying sedation. In either situation, the protocol dictates that the bedside nurse consults with the clinician on duty.

## Spontaneous breathing trial (SBT)

During weaning from mechanical ventilation, the BEACON Caresystem monitors the levels of Pressure Support (PS) and Positive End Expiratory Pressure (PEEP). When these levels are reduced to values below clinical guideline thresholds,[21] the BEACON Caresystem automatically monitors variables evaluating spontaneous breathing criteria. If these remain within limits and for a duration determined by international guidelines,[21] the system prompts the user to assess the appropriateness of extubation and provides a checklist to help. The primary function of the BEACON Caresystem with regards weaning is to reduce levels of support where possible; it does not suggest an active SBT or extubation. As such, the decision to extubate is according to standard clinical practice for both arms of the study, meaning that at any time the clinical team can perform an active SBT or extubate the patient without confirmation from the system. Reintubation is documented by the research team and the participant enters a standard care pathway, which involves the deactivation of the BEACON Caresystem. The participant is then followed up until the end of the protocol (figure 1).

The protocol ends with transfer or discharge from ICU or death (figure 1).

## Primary and secondary outcomes

The primary outcomes include:
▶ Duration of mechanical ventilation to successful extubation defined as the time from the start of mechanical ventilation, defined as either the time of intubation in the ICU, or the time of admission to the ICU following previous intubation for surgery, and until successful extubation, with successful extubation defined as ≥48 hours of unassisted spontaneous breathing after extubation.

Unassisted spontaneous breathing is defined here to include supplementary oxygen and support with high-flow nasal catheters. Patients with non-invasive mask ventilation are considered to be assisted.

The secondary objectives of the trial are to compare the following variables in the standard care group and BEACON Caresystem group:
▶ Duration of mechanical ventilation to successful extubation following randomisation defined as the time from randomisation and until successful extubation, with successful extubation defined as ≥48 hours of unassisted spontaneous breathing after extubation.
▶ Time from control mode to support mode defined as the time following randomisation, from initiation of control modes of ventilation and until initiation of support modes of ventilation.
▶ Time from support mode to successful extubation defined as the time following randomisation, from initiation of support modes of ventilation and until successful extubation.

▶ Number of changes in ventilator settings per day defined as the daily registered number of changes to ventilator settings from randomisation until successful extubation.
▶ Time to first SBT defined as the time from randomisation to the first performed SBT.
▶ Time to first successful SBT defined as the time from randomisation to the first successful SBT.
▶ Time to first extubation defined as the time from randomisation to the first extubation attempt.
▶ Time to first disconnection from mechanical ventilation defined as the time from randomisation to the point at which mechanical ventilation is first discontinued.
▶ Percentage of time in control mode ventilation defined as the time from randomisation which is spent in controlled modes of mechanical ventilation in per cent of duration of mechanical ventilation.
▶ Percentage of time in support mode ventilation defined as the time from randomisation which is spent in support modes of mechanical ventilation in per cent of duration of mechanical ventilation.
▶ Use of sedatives defined as the cumulative use of sedative drugs from randomisation until successful extubation.
▶ Use of neuromuscular blockading agents defined as the cumulative use of neuromuscular blockading agents from randomisation until successful extubation.
▶ Number of intubation free days defined as the number of days without intubation from randomisation until successful extubation.
▶ Number of reintubations defined as the number of reintubations following extubation from randomisation until successful extubation.
▶ Number of tracheostomies defined as the number of patients having tracheostomy performed from randomisation until successful extubation or protocol end.
▶ Number of patients on prolonged mechanical ventilation defined as the number of patients on mechanical ventilation ongoing for more than 21 days after the start of mechanical ventilation.
▶ Number and types of adverse events related to mechanical ventilation defined as the incidence of adverse events directly related to mechanical ventilation.
▶ ICU mortality defined as the mortality from randomisation and until death or ICU discharge.
▶ Length of stay defined as the duration of ICU admission from the start of mechanical ventilation and until death or ICU discharge.
▶ Mobilisation times defined as the time from randomisation until (1) first mobilisation, for example, sitting on the edge of the bed, standing up and marching on the spot and (2) regaining independency, for example, able to drink/eat or comb hair.
▶ Daily patient physiological blood gas status defined as daily ABG values of $SaO_2$, $PaO_2$, $PaCO_2$, pH from randomisation until successful extubation.

► Daily average physiological status defined as daily averages of measured values of oxygenation ($SpO_2$), end-tidal $CO_2$ fraction ($EtCO_2$), ventilation (respiratory rate, tidal volume, anatomical dead space), pulmonary mechanics (airway pressure, respiratory system compliance) and ventilator settings as measured routinely at the bedside.

## Statistical considerations

Sample size was determined by a power analysis of duration of mechanical ventilation. Extrapolation of the 2016 ICU admission data from two London-based NHS hospitals provided a sample size of 548 patients over an 18-month period. In these patients, mean total duration of mechanical ventilation measured in days was 7.2, with an SD of 7.0 days. A study population of 137 patients in each randomisation group would allow detection of a 2 day or 30% improvement in the duration of mechanical ventilation in the BEACON Caresystem group assuming a similar improvement in SD, with an α level of 0.05 and power of 0.8. A 2 days or 30% reduction in duration of mechanical ventilation is similar to that used previously.[6]

Primary and secondary outcomes will be measured across the protocol duration and comparison made by independent samples t-test, or appropriate non-parametric testing if not normally distributed. Complementary analysis will be performed using Kaplan-Meier curves.

The study is not sufficiently powered to allow subgroup analysis for the groups specified in the randomisation strategy.

An independent data safety monitoring committee will be established as part of the study, with this committee responsible for defining stopping criteria. Safety data will be looked at after the first 80 patients have been recruited, with an interim analysis performed following 50% recruitment.

## SAFETY, ETHICS AND DISSEMINATION
### Adverse events

All adverse events will be monitored and documented until the end of study. Adverse events are reported to the trial sponsor according to their standard operating procedure for further investigation. As patients are critically ill, it is expected that unrelated adverse events will occur as part of hospital stay. Adverse events are therefore only defined in relation to changes in mechanical ventilation, for patients allocated to either the control or intervention arms.

1. Serious adverse events are defined as:
   – Death, suspected to be caused directly by the settings of mechanical ventilation.
2. Adverse events include the following events suspected to be caused directly by the settings of mechanical ventilation:
   – Pneumothorax.
   – Self-extubation.

– Hypoxaemia ($SpO_2$ <90%; $PaO_2$ <8.00 kPa).
– Alkalaemia (pH >7.5).
– Acidaemia (pH <7.2).

## Safety considerations, systems advice

The potential risks and safety considerations related to the use of the the BEACON Caresystem have been considered and include:

### Inappropriate suggestions for changes in ventilator settings

Inappropriate changes in ventilator settings, if applied, can have serious consequences. The general strategy employed in this protocol has three components:

1. The BEACON Caresystem only provides advice in the situation where the models can accurately describe all psychological measurements. The ability of BEACON Caresystem to do so is automatically assessed by the system throughout the protocol and failure to do so means that no advice is provided, which is made visible to the user.
2. Maximal limits have been applied to the amount for which each ventilator setting can be changed in a single advice, meaning that inappropriate advice should have a reduced impact. The maximal advice changes for each ventilator setting are given in table 1.
3. The risks of changing each of the ventilator settings have been considered individually. Low values of inspiratory pressure or volume can result in reduction of minute ventilation and increase in $PaCO_2$ and consequently a fall in arterial pH. To assess this, $CO_2$ is monitored continuously by the BEACON Caresystem using capnography. High values of inspiratory pressure can result in lung damage. Peak inspiratory pressure is monitored continuously by the BEACON Caresystem, and all mechanical ventilators provide alarms at high values. Low values of respiratory frequency have no apparent detriment other than those specified in reduction of volume. High values of respiratory frequency can result in air trapping. The maximum step in respiratory frequency is limited to 3 breaths/min if the current frequency is ≥18 allowing the onset of trapping to be identified clinically.
4. Although there are small long-term risks associated with increasing inspiratory oxygen due to absorption atelectasis or oxygen toxicity, the major acute risk of changes in inspired oxygenation is arterial hypoxaemia. If the current $SpO_2$ ≤92%, then the maximum reduction in $FiO_2$ is limited to 2%. In addition, pulse oximetry is monitored continuously by the BEACON Caresystem during the study and an alarm sounded if levels are ≤90%.

To encourage use of the system and perform an evaluation where advice is applied, the application of advice is mandatory. As clinical practice is variable and advice is unlikely to be applied in all cases, the application and effects of advice are monitored by the BEACON Caresystem, and can be interrogated to ensure that the proposed changes have been made and that the effects are considered safe in cases of doubt.

## System malfunction

In the event of system malfunction, the patient enters standard care until the malfunction is detected and corrected by biomedical engineers.

## Sampling and measurement of ABG values

ABG sampling is part of routine ICU care. Additional ABG sampling may be required by the BEACON Caresystem than those performed in clinical practice. The BEACON Caresystem requirements, estimated from current experience, are expected to be a maximum of 6 per day in the intervention group.

All other measurements used are obtained from non-invasive means including pulse oximetry and measurement of respiratory gas content, flow and pressure.

## Ethical approval and consent

Study approval was obtained from the regional ethics committee of the HRA, REC reference: 17/LO/0887. IRAS reference: 226 610. The majority of the trial population lack the capacity to consent. Therefore, participant information sheets and advice are requested from a third party acting as a consultee; in most cases, this person will be a personal consultee, who is someone who knows the person lacking capacity and is able to advise the researcher about that person's wishes and feelings in relation to the project and whether they should join the research. However, where the personal consultee is not available, the researcher may nominate a professional person to assist in determining the participation of a person who lacks capacity. A nominated professional consultee will be appointed by the researcher to provide advice regarding the patient's suitability to participation in the research. In the event that a personal consultee is identified after the nominated professional consultee advice has been obtained, the above process for personal consultee declaration will be followed. A patient information sheet will be distributed immediately following the patient being identified as eligible for the study. If a surviving patient regains capacity, they will be approached by the research team to obtain their consent to continue in the study. If the patient refuses consent or withdraws, the intervention will be stopped but the regular/expected medical care will still be provided. Without consent, no additional information about the patient will be collected for the purposes of the study. However, to maintain integrity of the randomised trial, all information collected up to that time will still be used and analysed as part of the study.

## Dissemination plan

The dissemination strategy will involve the presentation of the results at international critical care conferences and publication in a peer-reviewed journal.

## Patient and public involvement

Patients and the public were not involved in the design of this study. Patients randomised into the study will be informed of the results of the study once the outcome of the study is published.

## DISCUSSION

This paper presents the study protocol of single-blinded multicentre randomised control trial evaluating management of mechanical ventilation as directed by the BEACON Caresystem compared with that of standard care in the general intensive care setting. To test the hypothesis that following advice from the BEACON Caresystem will reduce time to successful extubation, patients are randomised to care including advice from the system, or standard care. In addition, as secondary outcomes, the study will investigate whether application of the advice changes the time spent in control ventilation modes, the number of changes made to the ventilator, the use of medication such as sedative or neuromuscular blocking agents, the changes in clinical procedures such as reintubation or tracheostomy and clinical and physiological status of the patient. Although not powered to detect differences between groups, it will also investigate the length of stay at the ICU and patient mortality.

This study is the next step in evaluation of the advice of the BEACON Caresystem. A large number of studies have been performed validating the physiological models included in the system, as summarised in,[9] and studies of short duration have been performed evaluating the physiological response to application of advice.[14 15] This use of the system over the whole duration of ICU stay allows for assessment of more meaningful clinical outcomes, relating to the time spent on mechanical ventilation. In addition, the study includes only patients intubated for a minimum of 24 hours, meaning that short-term postoperative patients who may have rather uncomplicated intubation periods are not included.

The study includes limitations. The study was powered on duration of mechanical ventilation rather than duration to successful ventilation, this being the historical data available during study design. The study is not powered to detect changes in mortality, and does not follow-up on the patients after transfer from the ICU. In the context of this study, successful extubation is defined as the lack of need for non-invasive ventilator support for 48 hours after extubation, rather than successful rehabilitation of the patient subsequent to their ICU stay.

The results of this study compare care provided with the system's advice with current clinical practice at study sites. As there is no standard of care or usual method of weaning in the UK, this study does not compare the system's advice with that of best practice defined by clinical guidelines. To avoid comparing two interventional changes, no attempt will be made to standardise care in the control arm according to best practice.

As patients are studied for many days, it is not possible to store continuous waveform data for all patients at all times. Systematic analysis of patient-ventilator asynchrony will not therefore be possible.

This study excluded patients if ventilated for <24 hours. In doing so, patients ventilated post surgery defined previously as simple to wean are not included. This should not imply that the patients studied here are necessarily

difficult to wean, but rather that they represent a broad cross section of mechanically ventilated patients not simply recovering from the effects of anaesthesia.

In addition, it is important to note that substantial training has been provided to all sites in the use of the BEACON Caresystem and the same clinical team treat patients for both the control and intervention arm. It is possible then that both training and use of the system influence routine care, potentially biasing study results.

This paper describes the study protocol of a randomised control trial evaluating the advice of the BEACON Caresystem in a general medical ICU population. This protocol represents an important step evaluating the use of advice over the whole of the ICU stay, and only in patients intubated greater than 24 hours. The results will indicate whether the BEACON Caresystem is a useful tool in optimising ventilation strategy.

**Author affiliations**
[1]APMIC, Imperial College London, London, UK
[2]Magill Department of Anaesthesia and Intensive Care Medicine, Chelsea and Westminster Healthcare NHS Trust, London, UK
[3]The Respiratory and Critical Care Group (rcare), Department of Health Science and Technology, Aalborg University, Aalborg, Denmark
[4]Research Trial Unit, Chelsea and Westminster Hospital NHS Foundation Trust, London, UK
[5]Magill Department of Anaesthesia, Chelsea and Westminster Hospital NHS Foundation Trust, London, UK
[6]Anaesthetic Department, West Middlesex University Hospital NHS Trust, London, UK
[7]Anaesthetic Department, North Middlesex University Hospital NHS Trust, London, UK

**Acknowledgements** The authors would like to thank Sundhiya Mandelia, Senior Statistician, for reviewing the Method section of this protocol. Damon Foster and Essam Ramhamadany for their support and contributions to the setup of the clinical study. The critical care nurses and physicians working in the NHS who feedback on the protocol.

**Contributors** MPV, DSK and SR conceived, designed and wrote the protocol. MPV drafted, reviewed and amended the manuscript. LM, edited figures, drafted R&D documents. JRW drafted Standard operating procedure, created original figures and drafted the initial protocol manuscript. SS, AG, LO, LM, JM-C, SR and MPV reviewed and approved the final draft of the manuscript.

**Funding** This work was supported by Mermaid Care A/S and the National Institute of Health Research (NIHR).

**Competing interests** SR and DSK are minor shareholders of Mermaid Care A/S who manufacture the Beacon Caresystem. They have also acted as consultants for Mermaid Care A/S. SR is an unpaid board member of Mermaid Care A/S.

**Patient and public involvement** Patients and/or the public were not involved in the design, or conduct, or reporting, or dissemination plans of this research.

**Patient consent for publication** Not required.

**Provenance and peer review** Not commissioned; externally peer reviewed.

**ORCID iD**
M P Vizcaychipi http://orcid.org/0000-0001-7894-873X

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
