## [Reviewer comments · BMJ Open]

ARTICLE DETAILS

TITLE (PROVISIONAL)	The iCareWean protocol on weaning from mechanical ventilation: A Single-blinded multi-centre randomised control trial comparing an open-loop decision support system and routine care, in the general Intensive Care Unit
AUTHORS	Vizcaychipi, M; Martins, Laura; White, James; Karbing, Dan; Gupta, Amandeep; Singh, Suveer; Osman, Leyla; Moreno-Cuesta, Jeronimo; Rees, Steve

VERSION 1 – REVIEW

REVIEWER	Karen Burns St. Michael's Hospital, Toronto, Canada
REVIEW RETURNED	04-Jul-2019

GENERAL COMMENTS	General comments The research question the authors address is timely and topical. The research question is clear and the protocol is well written. The scientific context is described sufficiently to enable readers to understand the study design. The study methods are thoughtful but some questions remain and could be considered/addressed in the study methods, analysis, and discussion sections to enhance the quality of the trial and ultimately publication of the manuscript in the literature. Major concerns 1. Study Population: Although the background states that most studies have been conducted in simple to wean patients, the authors have not made an effort to identify a difficult to wean population or to characterize how their study population addresses this issue. This could be highlighted in the Discussion section. 2. Study Intervention: Control Arm Should a weaning protocol with daily screening to identify patients who are ready to undergo an SBT and suggestions for SBT techniques (T-piece, PS with PEEP) be used in the control arm as opposed to usual care? Current evidence supports that daily screening is the standard of care internationally. Although there is not a single SBT technique that has been convincingly demonstrated to be superior to another – a recent trial (Subira et al, JAMA), a systematic review (Burns et al, Critical Care) and guidance documents (ATS/ACCP guideline, Task Force on Weaning) provide evidence to guide clinicians on how to 'wean' patients from invasive ventilation and how to conduct 'SBTs'. Can the authors justify using Usual Care in the control arm of this
---

	trial? This is a major concern and should be addressed in the Discussion section. 3. Consistency of reporting and Selection of Primary Outcome: In the methods section of the abstract the authors imply that the primary outcome is duration of mv (alone). However, in the study design section of the protocol they allude to the primary outcome being time to successful extubation. Later in the protocol (under primary outcomes) they describe having 2 primary outcomes: both time to successful extubation and the duration of MV. A second issue is whether the investigators should prioritize ONE primary outcome – at present there are 2 primary outcomes specified in the protocol. Time to successful extubation was identified by a recent Delphi process to be the most important outcome to report in mechanical ventilation/weaning trials and has been identified as important to patients, family members, and clinicians. Skewness is a particular concern with duration of mechanical ventilation. Can the authors clarify the rationale for reporting 2 primary outcomes vs. prioritizing 1 primary outcome and including the other as the first secondary outcome? A third issue related to the primary outcome selection is the fact that the sample size estimation was based solely on one primary outcome (duration of MV). We do not know how much power they have to detect a difference in time to successful extubation. The primary outcome should be harmonized throughout the protocol. Although I realize that it may be too late to change the details of the trial protocol, I feel that these issues should be discussed openly in the Sample Size Estimation, Analysis, and Discussion sections of the protocol. 4. Stratification and Analysis: Given the size of the RCT (n=274), I am surprised to see that the authors wish to stratify patients based on 3 different patient features. Is there compelling evidence to support stratification? An alternative consideration may be to assess for heterogeneity of treatment effects after-the-fact. How will the authors take stratification into consideration in the analysis? Minor concerns 1. Background: The authors do not provide information on how often the iCareWean protocol make measurements and proposes changes? Can the ventilator log book be interrogated and examined to ensure that proposed changes were addressed 2. Study Intervention: iCareWean (intervention) arm Does PEEP greater than 5 cm H2O or a certain level prevent participants from transitioning into an SBT? What levels of PS/PEEP are SBTs conducted on? Does this vary based on the type of airway prosthesis (ETT tube vs. tracheostomy tube) or the type of humidification system being used (Heated Humidity vs Heat and Moisture Exchanger) in individual patients?
--	--

	3. Definition of time to successful extubation: How will the others consider use of high flow nasal cannulae (HFNC) in their definition of successful extubation?
--	---

REVIEWER	richard branson University of Cincinnati USA
REVIEW RETURNED	21-Jul-2019

GENERAL COMMENTS	Thank you for the description of the proposed study of evidence based guidance for ventilator management. I have a few concerns.  1. I take from your description that the system is capable of providing advice for both pressure and volume modes of ventilation. What modes are not able to be used ? APRV? etc. Is there any concern that in the cohort more patients will receive volume vs pressure ventilation which of course will alter the ability to limit VT? I believe it might be advisable to limit the subjects to volume of pressure ventilation to maintain consistency. 2. Under complications of ventilation you include self extubation. I fail to see ho the ventilator settings might effect this event. Sedation and restraints are far more important. Additionally I would include hypercarbia and hyperoxia as events. The impact of hyperoxia on outcome has become more evident in recent years. 3. Perhaps the glaring data that is missing is asynchrony. Asynchrony should be assessed in the two groups and would be an important outcome variable. Automated systems are available to detect types of asynchrony and asynchrony index. 4. It is likely difficult to include the operation of the BEACON system in this paper describing the trial - but some more detail would be helpful 5. I would also warn that engineers often see the ability of automated or decision support systems to reduce clinician interaction as an advantage. In this case - the 'awareness' of caregivers should be assured. Less interaction between the caregivers and the device is not necessarily a step forward. 6. What are the rules for clinicians ability to overrule the advice of the BEACON system?
--

REVIEWER	Craig Jabaley Emory University Atlanta, GA, USA
REVIEW RETURNED	29-Jul-2019

GENERAL COMMENTS	The authors have submitted a trial protocol for the comparison of software-driven weaning and separation from mechanical ventilation compared to standard care. This work will build on prior published work by the authors examining the clinical impact of the studied software platform, which can provide clinical guidance about the maintenance, weaning, and discontinuation of mechanical ventilation. On balance the protocol is well-presented and relatively complete. I have concerns about a few weaknesses inherent to the trial design that should be acknowledged and addressed. Otherwise, additional clarity around a few points would likely benefit readers who are unfamiliar with this software platform and prior work published by the authors. --Major considerations:  1. Methods, page 5, general comment: The authors have not provided much background as to what comprises "standard care." Is
---

	this governed by protocols, either as part of the trial or as part of routine local practice? If the former, these should be included as supplements. If the latter, additional information is needed to set the stage for readers. For example, are there standard and widely utilized protocols for the weaning and separation from mechanical ventilation? Is there a routine frequency with which patients are screened and evaluated for SBT readiness? Looking ahead, there are related potential and valid criticisms of the trial. Might any observed differences in outcome be due to solely to increased attention to patients in the intervention arm? 2. Methods, page 7, study procedure: Similarly, another potential criticism is the extent to which clinicians expect or reject the software platform's advice. One point that is not clear from the present manuscript, but can be inferred from the authors' related work, is that clinicians are free to reject the advice of the system. This should be clarified. Will this outcome be tracked and reported? If so, it does not appear in the outcomes section. Additionally, this could be included in the section concerning safety considerations as final clinician review does serve as a backstop against changes determined to be potentially unsafe. 3. Methods, page 8, intervention arm, lines 7-17: This section would benefit from expansion and further clarity. The time and/or parameter (i.e., RSBI) thresholds where the system recommends either a SBT or extubation should be delineated in the manuscript or in a supplement. Additionally, elements of the referenced extubation checklist should be further delineated. 4. Methods, page 8, primary & secondary outcomes: How will the authors handle clinical variability in the selection of, and reporting of, non-invasive ventilation support after extubation? This could include non-invasive positive pressure ventilation or heated high-flow nasal cannula. 5. Methods, page 9, statistical considerations, line 56: Are there predefined safety or other stopping points for the trial? Is there a formal interim analysis plan? --Other considerations: 6. Introduction, page 4, lines 32-33: The term "more severe" is ambiguous. It could be argued that the trial's exclusion criteria will prevent enrollment of severe compromised patients, for example. I would specify here and in related areas throughout the manuscript that the goal seems to be recruitment of patients who have mechanical ventilation needs more than a few hours of "routine" postoperative intubation. 7. Methods, page 5, general comment: Optionally, I would consider moving the section on ethics earlier into the methods as this organizational structure may be more familiar to readers. 8. Methods, page 5, administrative structure: Please include the planned dates of the trial. 9. Methods, page 5, description of the BEACON Caresystem: The inclusion of graphical depictions of the software interface and/or other visual representations of its underlying approach would likely be helpful for unfamiliar readers. These could be adapted from prior publications by this group and included as appendices. 10. Methods, page 6, description of the BEACON Caresystem, Lines 11-12: As a minor point, the abbreviation PEEP occurs before its full expansion on the subsequent line. 11. Methods, page 6, eligibility criteria, lines 32-34: Can the authors clarify their exclusion criterion related to the use of vasopressors? Does any amount of vasopressor lead to exclusion? The authors list a norepinephrine cutoff – are norepinephrine equivalents to be
--	---

	calculated for other vasopressors? 12. Methods, page 7, intervention arm, line 47: The acronym ALPE, presumably for automatic lung parameter estimator, needs a bit more explanation here for readers who are not familiar with related work. 13. Safety, Ethics, and Dissemination, page 10, adverse events, line 23: "pH > 7.2" should be corrected to "pH < 7.2"
--	--

VERSION 1 – AUTHOR RESPONSE

Reviewer: 1

Reviewer Name

Karen Burns

Institution and Country

St. Michael's Hospital, Toronto, Canada

Please state any competing interests or state 'None declared':
I am the Principal Investigator a North American weaning trial.

Please leave your comments for the authors below

Peer Review – BMJ Open

Manuscript ID bmjopen-2019-031813,

Title: "The iCareWean protocol on weaning from mechanical ventilation: A Single-blinded multi-centre randomised control trial comparing an open-loop decision support system and routine care, in the general Intensive Care Unit.."

Reviewed by: Dr. Karen E. A. Burns

General comments

The research question the authors address is timely and topical. The research question is clear and the protocol is well written. The scientific context is described sufficiently to enable readers to understand the study design. The study methods are thoughtful but some questions remain and could be considered/addressed in the study methods, analysis, and discussion sections to enhance the quality of the trial and ultimately publication of the manuscript in the literature.

Major concerns

1. *Study Population: Although the background states that most studies have been conducted in simple to wean patients, the authors have not made an effort to identify a difficult to wean population or to characterize how their study population addresses this issue. This could be highlighted in the Discussion section.*

Thank you for useful comments. It is clear that our definition was misleading, and the following text has been added to the descriptions of limitations in the discussion.

“This study excluded patients if ventilated for <24 hours. In doing so, patients ventilated post-surgery defined previously as “simple to wean” are not included. This should not imply that the patients studied here are necessarily difficult to wean, but rather that they represent a broad cross-section of mechanically ventilated patients not simply recovering from the effects of anaesthesia.”

--

2. Study Intervention: Control Arm

Should a weaning protocol with daily screening to identify patients who are ready to undergo an SBT and suggestions for SBT techniques (T-piece, PS with PEEP) be used in the control arm as opposed to usual care?

Current evidence supports that daily screening is the standard of care internationally. Although there is not a single SBT technique that has been convincingly demonstrated to be superior to another – a recent trial (Subira et al, JAMA), a systematic review (Burns et al, Critical Care) and guidance documents (ATS/ACCP guideline, Task Force on Weaning) provide evidence to guide clinicians on how to ‘wean’ patients from invasive ventilation and how to conduct ‘SBTs’.

Can the authors justify using Usual Care in the control arm of this trial? This is a major concern and should be addressed in the Discussion section.

Thank you for your suggestion and sharing references regarding available data in the literature. Unfortunately, there is not a ‘standard and/or usual’ recommended method of weaning in clinical practice in the UK and we decided to compare the system to current clinical practice. We recognise the limitations of not comparing against standardized ‘best’ practice, however our concern was that by standardising the control arm of the study we in effect introduce two interventional changes to current practice at the sites. The following text has been added to the discussion.

“The results of this study compare care provided with the system’s advice with current clinical practice at study sites. As there is no standard of care or usual method of weaning in the United Kingdom, this study does not compare the system’s advice with that of best practice defined by clinical guidelines. To avoid comparing two interventional changes, no attempt will be made to standardize care in the control arm according to best practice.”

3. Consistency of reporting and Selection of Primary Outcome: *In the methods section of the abstract the authors imply that the primary outcome is duration of mv (alone). However, in the study design section of the protocol they allude to the primary outcome being time to successful extubation. Later in the protocol (under primary outcomes) they describe having 2 primary outcomes both time to successful extubation and the duration of MV.*

A second issue is whether the investigators should prioritize ONE primary outcome – at present there are 2 primary outcomes specified in the protocol. Time to successful extubation was identified by a recent Delphi process to be the most important outcome to report in mechanical ventilation/weaning trials and has been identified as important to patients, family members, and clinicians. Skewness is a particular concern with duration of mechanical ventilation. Can the authors clarify the rationale for

reporting 2 primary outcomes vs. prioritizing 1 primary outcome and including the other as the first secondary outcome?

A third issue related to the primary outcome selection is the fact that the sample size estimation was based solely on one primary outcome (duration of MV). We do not know how much power they have to detect a difference in time to successful extubation.

The primary outcome should be harmonized throughout the protocol. Although I realize that it may be too late to change the details of the trial protocol, I feel that these issues should be discussed openly in the Sample Size Estimation, Analysis, and Discussion sections of the protocol.

Thank you for your useful comments. We apologise for this discrepancy. The primary outcome is the duration of mechanical ventilation until successful extubation defined as ≥ 48 hours unassisted spontaneous breathing after extubation. This has now been made consistent throughout the manuscript, and the definition of unassisted spontaneous ventilation clarified. Unassisted spontaneous breathing is defined here to include supplementary oxygen and support with high flow nasal catheters. Patients with non-invasive mask ventilation are considered to be assisted, as specified in the definition of outcomes.

The reviewer is correct that power analysis was performed on duration of mechanical ventilation, rather than duration to successful extubation, as this was the historical data available to us during study design. This has been included as a limitation in the discussion section

4. Stratification and Analysis: Given the size of the RCT (n=274), I am surprised to see that the authors wish to stratify patients based on 3 different patient features. Is there compelling evidence to support stratification? An alternative consideration may be to assess for heterogeneity of treatment effects after-the-fact. How will the authors take stratification into consideration in the analysis?

Thank you for your comments. The purpose of stratification and block randomisation was to increase the probability that patients with particular diagnoses were distributed evenly between the control and intervention groups. We had no compelling evidence that different diagnoses should result in different outcome, however we felt that to minimise the risk of potential bias it was important to stratify in this way. The study is not powered to investigate changes in primary outcome between groups.

The following sentence has been modified in the 'Randomisation' section.

"block randomisation is applied with each block stratified to ensure equal distribution between control and intervention arm for the following patient groups, reducing the risk of potential bias by overrepresentation of these groups in the control or intervention arms:"

Minor concerns

1. Background:

The authors do not provide information on how often the iCareWean protocol make measurements and proposes changes?

Can the ventilator log book be interrogated and examined to ensure that proposed changes were addressed

We apologise for this omission. Thank you for highlighting it. The following text has been added to the description of the intervention arm.

“New advice is generated only when the system regards current settings as not appropriate. Duration between advice therefore varies, however for PEEP and changes in volume or pressure 10-20 minutes is the minimum period between advice.

The Beacon Care system records all advice, changes and responses, effectively functioning as a ventilator log book. The following text has been added to the safety considerations related to patient advice to explain this.

“The application and effects of advice are monitored by the Beacon Caresystem, and can be interrogated to ensure that proposed changes have been made and that the effects are considered safe in cases of doubt.”

2. Study Intervention: iCareWean (intervention) arm

Does PEEP greater than 5 cm H₂O or a certain level prevent participants from transitioning into an SBT?

What levels of PS/PEEP are SBTs conducted on? Does this vary based on the type of airway prosthesis (ETT tube vs. tracheostomy tube) or the type of humidification system being used (Heated Humidity vs Heat and Moisture Exchanger) in individual patients?

The Beacon system functions so as to reduce mechanical ventilation where possible. For the patient in pressure support mode, this means that the system will suggest a reduction in PS where the patient does not increase the RR/Vt ratio to elevated levels and where the measured oxygen consumption does not become elevated following PS reduction. The primary function of the system in this regard is to wean the patient to the point where SBT and extubation might be considered clinically beneficial. The system is not designed to suggest an active SBT procedure, or to suggest extubation. The only functionality provided in this regard is to monitor the patient when values of PEEP and PS are below threshold values determined from guidelines [21], providing a counter which counts down from 30 minutes, and as such indicates that the patient has been within ventilator settings consistent with an SBT and stable with regard other respiratory measurements, for this duration. In the protocol, this situation triggers the nurse to review the extubation checklist and contact the doctor responsible for the patient to discuss extubation. It does not however, limit current practice in relation to extubation, with decisions made according to standard clinical practice. The Beacon system does not therefore prevent participants from transitioning to active SBT or extubation if this is the decision of the clinical team.

The text describing SBT in the intervention arm has been modified to highlight this with the text now reading.

“During weaning from mechanical ventilation the BEACON Caresystem© monitors the levels of PS and PEEP. When these levels are reduced to values below clinical guideline thresholds [21], the BEACON Caresystem© automatically monitors variables evaluating spontaneous breathing criteria. If these remain within limits and for a duration determined by international guidelines [21], the system prompts the user to assess the appropriateness of extubation and provides a checklist to help. The primary function of the BEACON Caresystem© with regards weaning is to reduce levels of support where possible, it does not suggest an active SBT or extubation. As such, the decision to extubate is according to standard clinical practice for both arms of the study, meaning that at any time the clinical team can perform an active SBT or extubate the patient without confirmation from the system. Reintubation is documented by the research team and the participant enters a standard care pathway, which involves the deactivation of the BEACON Caresystem©. The participant is then followed up until the end of the protocol (Figure 1).”

3. Definition of time to successful extubation: How will the others **[authors]** consider use of high flow nasal cannulae (HFNC) in their definition of successful extubation?

Thank you for your comments. The definition of unassisted spontaneous ventilation has now been clarified in the description of the primary outcome. Here we have added the text

“Unassisted spontaneous breathing is defined here to include supplementary oxygen and support with high flow nasal catheters. Patients with non-invasive mask ventilation are considered to be assisted.”

Reviewer: 2

Reviewer Name

richard branson

Institution and Country

University of Cincinnati
USA

Please state any competing interests or state ‘None declared’:
none

Please leave your comments for the authors below Thank you for the description of the proposed study of evidence based guidance for ventilator management. I have a few concerns.

1. I take from your description that the system is capable of providing advice for both pressure and volume modes of ventilation. What modes are not able to be used ? APRV? etc. Is there any concern that in the cohort more patients will receive volume vs pressure ventilation which of course will alter the ability to limit VT? I believe it might be advisable to limit the subjects to volume of pressure ventilation to maintain consistency.

Thank you for reviewing our manuscript and providing useful comments. The system provides advice in both pressure and volume modes, and also for mandatory and spontaneous breathing. The system does not currently support APRV, or modes that are in themselves advising on or regulating ventilation such as SmartCare or ASV. No attempt has been made to limit the modes used as we wish to follow the patient during their whole course of mechanical ventilation, regardless as to the mode selected. The system does however advise on switching between mandatory and spontaneous breathing modes, as described in the manuscript. As the system does not suggest whether a control or pressure mode should be applied – this is a clinical decision - there is no reason to believe that the study arms will be biased in terms of ability to regulate tidal volume. To understand the regulation of tidal volume the system records both the set and measured values of tidal volume, which allows identification of double triggering and excess inspiratory muscle activity during mandatory inspiration in volume control modes.

Electronic supplementary material has been added to the manuscript submission, describing more details of the functionality of the system. For clarity the following sentence was added to this ESM “data is collected automatically by the Beacon Caresystem, and includes tidal volumes, both set if the mode dictates, and also measured. The regulation of Vt and it relationship with mode can therefore be explored for both of the study arms”.

--

2. Under complications of ventilation you include self extubation. I fail to see ho the ventilator settings might effect this event. Sedation and restraints are far more important. Additionally I would include hypercarbia and hyperoxia as events. The impact of hyperoxia on outcome has become more evident in recent years.

Thank you for valuable comments. You are completely correct that self-extubation is unrelated to ventilator settings. We also agree about hypercarbia and hyperoxia.

As self-extubation is a part of the approved protocol, it would be difficult for us to request the ethics committee that this be removed. We hope that you understand this and that we have not modified the manuscript in this regard. With regard hyperoxia and hypercarbia, we record ventilator settings, capnography, SpO2 and all blood gas values, meaning that hypercarbia and hyperoxia can easily be compared in the study arms. As the protocol is already ethically approved we cannot modify this manuscript, however we intend to submit an amendment to the ethics committee to request that we may evaluate these aspects. Thank you for your suggestion.

--

3. Perhaps the glaring data that is missing is asynchrony. Asynchrony should be assessed in the two groups and would be an important outcome variable. Automated systems are available to detect types of asynchrony and asynchrony index.

This would be a very interesting sub-analysis to perform and we are aware that there are available tools for doing so. The Beacon system has the ability to collect waveform data, however doing so is limited by the storage requirements of the tablet on which the system is stored. As patients are studied over many days continuous waveform data has not been collected systematically for all patients over all periods. It may be possible to review cases of asynchrony in both arms on completion of data collection, but a systematic comparison of the nature of asynchrony in the study arms will not be possible. We are convinced that there is substantial information related to asynchrony in comparison of variables such as the measured and set tidal volume. We intend to explore this for inspiration, but commenting on this in the manuscript would be at best anecdotal. We have included this as a limitation in the discussion, adding the following text.

“As patients are studied for many days it is not possible to store continuous waveform data for all patients at all times. Systematic analysis of patient-ventilator asynchrony will not therefore be possible.”

4. It is likely difficult to include the operation of the BEACON system in this paper describing the trial - but some more detail would be helpful.

We have included a small supplementary material illustrating the function of the Beacon system. We hope this is helpful in illustrating its use in the study.

5. I would also warn that engineers often see the ability of automated or decision support systems to reduce clinician interaction as an advantage. In this case - the 'awareness' of caregivers should be assured. Less interaction between the caregivers and the device is not necessarily a step forward.

Thank you for your comment and reinforcement of the concept of caregiver interaction with equipment. This is the main difference between an open and close loop system. The BEACON system aims to empower nurses and physicians to interact with the ventilator, hopefully benefitting patients. In brief, the open-loop nature of the Beacon system means that advice is not automatically applied. In addition, it is based on a graphical visualisation of patient state, and access to a physiological description of the individual patient tuned to clinical data. It is our hope that this promotes bedside interaction, can be used as a bedside teaching tool, and encourages physiological thinking at the bedside. None of this is relevant for the manuscript, but I hope it is clear we share and agree with your concerns.

6. What are the rules for clinicians ability to overrule the advice of the BEACON system?

Thank you for this useful question. To test the system properly, i.e. encourage application of the advice, it was decided that following the advice was mandatory. This decision was made in full recognition that it was unlikely to be achieved in all cases. The Beacon system records all advice and all ventilator changes, meaning that a full analysis of acceptance of advice can be made. Please note, that in the event a clinician requires to intervene e.g. suctioning, tracheostomy, transfer the patient to radiology etc, then they can pause the BEACON system to perform the required procedure. This is a safe approach to allow clinicians to work dynamically with the patients. The following text has been added to the safety considerations.

“To encourage use of the system and perform an evaluation where advice is applied, the application of advice is mandatory. As clinical practice is variable and advice is unlikely to be applied in all cases, the application and effects of advice are monitored by the Beacon Caresystem, and can be interrogated to ensure that proposed changes have been made and that the effects are considered safe in cases of doubt.”

Reviewer: 3

Reviewer Name

Craig Jabaley

Institution and Country

Emory University
Atlanta, GA, USA

Please state any competing interests or state ‘None declared’:
None declared

Please leave your comments for the authors below

The authors have submitted a trial protocol for the comparison of software-driven weaning and separation from mechanical ventilation compared to standard care. This work will build on prior published work by the authors examining the clinical impact of the studied software platform, which can provide clinical guidance about the maintenance, weaning, and discontinuation of mechanical ventilation. On balance the protocol is well-presented and relatively complete. I have concerns about a few weaknesses inherent to the trial design that should be acknowledged and addressed. Otherwise, additional clarity around a few points would likely benefit readers who are unfamiliar with this software platform and prior work published by the authors.

--Major considerations:

1. *Methods, page 5, general comment: The authors have not provided much background as to what comprises “standard care.” Is this governed by protocols, either as part of the trial or as part of routine local practice? If the former, these should be included as supplements. If the latter, additional*

information is needed to set the stage for readers. For example, are there standard and widely utilized protocols for the weaning and separation from mechanical ventilation? Is there a routine frequency with which patients are screened and evaluated for SBT readiness? Looking ahead, there are related potential and valid criticisms of the trial. Might any observed differences in outcome be due to solely to increased attention to patients in the intervention arm?

Thank you for your comments. Similar concerns were raised by reviewer 1, and much of our response is copied here.

Unfortunately, there is not a 'standard and/or usual' recommended method of weaning in clinical practice in the UK and we decided to compare the system to current clinical practice. We recognise the limitations of not comparing against standardized 'best' practice, however our concern was that by standardising the control arm of the study we in effect introduce two interventional changes to current practice at the sites. The following text has been added to the discussion.

"The results of this study compare care provided with the system's advice with current clinical practice at study sites. As there is no standard of care or usual method of weaning in the United Kingdom, this study does not compare the system's advice with that of best practice defined by clinical guidelines. To avoid comparing two interventional changes, no attempt will be made to standardize care in the control arm according to best practice."

We do not believe that there should be differences in outcome due to solely to increased attention to patients in the intervention arm. Both control and intervention arms of the study have the Beacon system attached, and the bedside nurse is required to monitor the data to make sure that all data is correctly collected and that sensors – respiratory and pulse oximetry – are correctly placed. Our anecdotal observations in this regard have not been included in the manuscript.

2. Methods, page 7, study procedure: Similarly, another potential criticism is the extent to which clinicians expect or reject the software platform's advice. One point that is not clear from the present manuscript, but can be inferred from the authors' related work, is that clinicians are free to reject the advice of the system. This should be clarified. Will this outcome be tracked and reported? If so, it does not appear in the outcomes section. Additionally, this could be included in the section concerning safety considerations as final clinician review does serve as a backstop against changes determined to be potentially unsafe.

Thank you for your comments. To test the system properly, i.e. encourage application of the advice, it was decided that following the advice was mandatory. This decision was made in full recognition that it was unlikely to be achieved in all cases. The Beacon system records all advice and all ventilator changes, meaning that a full analysis of acceptance of advice can be made. The following text has been added to the safety considerations.

"To encourage use of the system and perform an evaluation where advice is applied, the application of advice is mandatory. As clinical practice is variable and advice is unlikely to be applied in all cases, the application and effects of advice are monitored by the Beacon Caresystem, and can be interrogated to ensure that proposed changes have been made and that the effects are considered safe in cases of doubt."

3. Methods, page 8, intervention arm, lines 7-17: This section would benefit from expansion and further clarity. The time and/or parameter (i.e., RSBI) thresholds where the system recommends either a SBT or extubation should be delineated in the manuscript or in a supplement. Additionally, elements of the referenced extubation checklist should be further delineated.

The Beacon system functions so as to reduce mechanical ventilation where possible. For the patient in pressure support mode, this means that the system will suggest a reduction in PS where the patient does not increase the RR/Vt ratio to elevated levels and where the measured oxygen consumption does not become elevated following PS reduction. The primary function of the system in this regard is to wean the patient to the point where SBT and extubation might be considered clinically beneficial. The system is not designed to suggest an active SBT procedure, or to suggest extubation. The only functionality provided in this regard is to monitor the patient when values of PEEP and PS are below threshold values determined from guidelines [21], providing a counter which counts down from 30 minutes, and as such indicates that the patient has been within ventilator settings consistent with an SBT and stable with regard other respiratory measurements, for this duration. In the protocol, this situation triggers the nurse to review the extubation checklist and contact the doctor responsible for the patient to discuss extubation. It does not however, provide advice related to performing an active SBT or extubation, with decisions on these made according to standard clinical practice.

The SBT set up and evaluation screens and extubation checklist are provided in the supplementary material for reference. The text describing SBT in the intervention arm has been modified to highlight the role of the Beacon system in relation to SBT and extubation, as follows

“During weaning from mechanical ventilation the BEACON Caresystem© monitors the levels of PS and PEEP. When these levels are reduced to values below clinical guideline thresholds [21], the BEACON Caresystem© automatically monitors variables evaluating spontaneous breathing criteria. If these remain within limits and for a duration determined by international guidelines [21], the system prompts the user to assess the appropriateness of extubation and provides a checklist to help. The primary function of the the BEACON Caresystem© with regards weaning is to reduce levels of support where possible, it does not suggest an active SBT or extubation. As such, the decision to extubate is according to standard clinical practice for both arms of the study, meaning that at any time the clinical team can perform an active SBT or extubate the patient without confirmation from the system. Reintubation is documented by the research team and the participant enters a standard care pathway, which involves the deactivation of the BEACON Caresystem©. The participant is then followed up until the end of the protocol (Figure 1).”

4. Methods, page 8, primary & secondary outcomes: How will the authors handle clinical variability in the selection of, and reporting of, non-invasive ventilation support after extubation? This could include non-invasive positive pressure ventilation or heated high-flow nasal cannula.

Thank you for your comments. The definition of unassisted spontaneous ventilation has now been clarified in the description of the primary outcome. Here we have added the text

“Unassisted spontaneous breathing is defined here to include supplementary oxygen and support with high flow nasal catheters. Patients with non-invasive mask ventilation are considered to be assisted.”

5. *Methods, page 9, statistical considerations, line 56: Are there predefined safety or other stopping points for the trial? Is there a formal interim analysis plan?*

The following text has been added to the section on statistical considerations

“An independent data safety monitoring committee will be established as part of the study, with this committee responsible for defining stopping criteria. Safety data will be looked at after the first 80 patients have been recruited, with an interim analysis performed following 50% recruitment.”

--Other considerations:

6. *Introduction, page 4, lines 32-33: The term “more severe” is ambiguous. It could be argued that the trial’s exclusion criteria will prevent enrollment of severe compromised patients, for example. I would specify here and in related areas throughout the manuscript that the goal seems to be recruitment of patients who have mechanical ventilation needs more than a few hours of “routine” postoperative intubation.*

Thank you for useful comments. The reviewer is completely correct and we have modified the manuscript accordingly. The introduction has been modified to address the specific comment of the reviewer, with the text now reading

“Consequently, it remains to be investigated whether decision support systems can reduce duration of mechanical ventilation in patients with more than a few hours of “routine” postoperative intubation. “

In addition the following text has been added to the limitations in response to a similar comment from another reviewer

“This study excluded patients if ventilated for <24 hours. In doing so, patients ventilated post-surgery defined previously as “simple to wean” are not included. This should not imply that the patients studied here are necessarily difficult to wean, but rather that they represent a broad cross-section of mechanically ventilated patients not simply recovering from the effects of anaesthesia.”

7. *Methods, page 5, general comment: Optionally, I would consider moving the section on ethics earlier into the methods as this organizational structure may be more familiar to readers.*

Thank you for your suggestion. We have now moved the section on ethics earlier into the methods.

-

8. *Methods, page 5, administrative structure: Please include the planned dates of the trial.*

Sorry for this omission. The planned dates for the study have now been added. September 2017 – Nov 2020

9. *Methods, page 5, description of the BEACON Caresystem: The inclusion of graphical depictions of the software interface and/or other visual representations of its underlying approach would likely be helpful for unfamiliar readers. These could be adapted from prior publications by this group and included as appendices.*

Thank you for highlighting this. A pictorial representation of the BEACON system was made and has been added to the electronic supplementary material.

10. *Methods, page 6, description of the BEACON Caresystem, Lines 11-12: As a minor point, the abbreviation PEEP occurs before its full expansion on the subsequent line.*

Sorry, suggested change has now been made.

11. *Methods, page 6, eligibility criteria, lines 32-34: Can the authors clarify their exclusion criterion related to the use of vasopressors? Does any amount of vasopressor lead to exclusion? The authors list a norepinephrine cutoff – are norepinephrine equivalents to be calculated for other vasopressors?*

Cardiovascular instability is one of the criteria that precludes clinicians from liberating patients from mechanical ventilation hence the need to introduce a definition of 'stability' for the purpose of standardisation inclusion criteria. As it is specified in the inclusion criteria list a dose of noradrenaline > 0.2mcg/Kg/min will be an exclusion criterion as it has to be below this dose to be included. In the UK, noradrenaline is the vasopressor of choice therefore there is no need to convert the dose. Thanks for comments.

12. *Methods, page 7, intervention arm, line 47: The acronym ALPE, presumably for automatic lung parameter estimator, needs a bit more explanation here for readers who are not familiar with related work.*

Thanks you for this comment. The acronym has now been spelt out, along with a reference to the electronic supplementary material, which provides further information.

13. *Safety, Ethics, and Dissemination, page 10, adverse events, line 23: “pH > 7.2” should be corrected to “pH < 7.2”*

Thank you for highlighting this. It has been modified as requested.

VERSION 2 – REVIEW

REVIEWER	Craig Jabaley Emory University Atlanta, GA, USA
REVIEW RETURNED	14-Oct-2019

GENERAL COMMENTS	With thanks to the authors, my prior concerns have been adequately addressed. The manuscript has improved from its prior state and now adequately describes the planned/ongoing trial, including delineation of its procedures and strengths/weaknesses.
--

REVIEWER	Luca Salvatore De Santo University of Campania, Naples, Italy
REVIEW RETURNED	18-Nov-2019

GENERAL COMMENTS	Nice research work.
---------------------

REVIEWER	Savino Spadaro UNIVERSITA' DI FERRARA
REVIEW RETURNED	17-Dec-2019

GENERAL COMMENTS	In this study protocol, the authors describe a single-blinded multi-centre randomised control trial evaluating management of mechanical ventilation guided by the BEACON Caresystem compared to that of standard care in the general intensive care setting. The study protocol is well described and detailed in all sections and overall is clear. The sample size is adequate to identify correctly the impact of this strategy. I have the following suggestions for the authors: Please clarify the characteristics of patients populations that will be included in the study and furthermore identify the subgroup analysis. The inclusion criteria include a Haemodynamically stable condition; this aspect should be clarified. In ICU, it is very frequent an unstable hemodynamic condition that could be limited the enrolment e could not reflect adequately the severity of critically ill patients.
---

	Several outcomes will test for this study. I suggest to collect the data regarding the hours of high flow nasal cannula and the flows used. How do you evaluate the timing of the mechanical ventilation? Do you include the duration of the use of high flow nasal cannula in the time of MV? A minimum of PEEP is offered by this system. How do you consider the use of NIV after extubation? Is it an adverse event? What are the criteria used for the extubation in the routine care group?
--	--

VERSION 2 – AUTHOR RESPONSE

Reviewer: 3

Reviewer Name

Craig Jabaley

Institution and Country

Emory University

Atlanta, GA, USA

Please state any competing interests or state 'None declared':

None declared

Please leave your comments for the authors below

With thanks to the authors, my prior concerns have been adequately addressed. The manuscript has improved from its prior state and now adequately describes the planned/ongoing trial, including delineation of its procedures and strengths/weaknesses.

A: Thank you for your comments.

Reviewer: 4

Reviewer Name

Luca Salvatore De Santo

Institution and Country

University of Campania, Naples, Italy

Please state any competing interests or state 'None declared':

none declared

Please leave your comments for the authors below

nice research work

A: Thank you for your comments.

Reviewer: 5

Reviewer Name

Savino Spadaro

Institution and Country

UNIVERSITA' DI FERRARA

Please state any competing interests or state 'None declared':

None declared

Please leave your comments for the authors below

in this study protocol, the authors describe a single-blinded multi-centre randomised control trial evaluating management of mechanical ventilation guided by the BEACON Caresystem compared to that of standard care in the general intensive care setting.

The study protocol is well described and detailed in all sections and overall is clear. The sample size is adequate to identify correctly the impact of this strategy.

A: Thank you for your comments.

I have the following suggestions for the authors:

Please clarify the *characteristics of patients* populations that will be included in the study and furthermore identify the *subgroup analysis*.

A: The **characteristics** of the patients are defined in the inclusion criteria (lines 55-59 in page 6 and lines 3-32 which also include exclusion criteria). In brief any patient admitted to the intensive care unit who required invasive mechanical ventilation for at least 24 hours. Regarding **subgroup analysis**. Three subgroups are used for randomisation, as described on p7 lines 45 to 48. Randomisation using these subgroups is intended to prevent bias due to overrepresentation as stated on p7 lines 42-43. The study is not, however, powered for subgroup analysis with effective subgroup randomisation allowing comparison of all patients included in the two arms. The following text has been added to the section 'statistical considerations' so as to be explicit

"The study is not sufficiently powered to allow subgroup analysis for the groups specified in the randomisation strategy"

The inclusion criteria include a Haemodynamically stable condition; this aspect should be clarified. In ICU, it is very frequent an unstable hemodynamic condition that could be limited the enrolment e could not reflect adequately the severity of critically ill patients.

Thank you for comment which we are agreed. We wanted this study to focus on general medical ICU patients, and not the most complex cardiovascular patients. In doing so we are aware of another study being undertaken so as to evaluate the system in a cardiothoracic ICU, so it seemed appropriate to limit this study.

In doing so we only exclude the most severe of these patients, recognising that mild circulatory compromise is a factor for many ICU patients. As such we do not believe this will seriously effect enrolment. The specific definition of 'haemodynamic instability used here is given on p7 lines 3-6, i.e. Haemodynamically stable, where instability is defined by the presence of two or more of the following criteria: acidosis pH < 7.2, poor urine output < 0.5ml/kg/h, use of vasopressors, e.g. noradrenaline > 25 µg/min."

Several outcomes will test for this study. I suggest collecting the data regarding the hours of high flow nasal cannula and the flows used. How do you evaluate the timing of the mechanical ventilation?

The hours and flow of the nasal catheter are included in the study design. These were mentioned on page 9 as outside out primary outcome, being considered unassisted ventilation, and as such it may have been unclear to the reader as to whether they were measured. They are also not included in our analysis of secondary outcomes. As the protocol is now approved we cannot add a secondary outcome to this manuscript. However, as the data is collected there may be a possibility to request approval for such an analysis at a later date, and as such your comment is valuable, even if it does not result in modification of the manuscript at this point. The timing of ventilation is as specified in our primary outcome and subsequent text on page 9.

Do you include the duration of the use of high flow nasal cannula in the time of MV? A minimum of PEEP is offered by this system.

Thank you for comments: As stated in the response to our previous answer, high flow nasal cannula is considered unassisted mode of mechanical ventilation in our protocol and not therefore a part of the duration included in the primary outcome, as stated on page 9. Indeed, it has been described the provision of additional level of PEEP (3-4cmH₂O) in paediatric population and to some extent in some clinical studies in adults but we feel the inclusion of high flow oxygen as part of invasive mechanical ventilation would not be appropriate in this study.

How do you consider the use of NIV after extubation? Is it an adverse event?

This is an active clinical decision therefore it is not included as adverse event but the information will be recorded as a clinical event. If within the 48 hours subsequent to extubation then it prolongs the period of the primary outcome, otherwise it is simply recorded as a clinical event. We believe that this is explicit in our primary outcome description on page 9.

What are the criteria used for the extubation in the routine care group?

Thank you for your question. Routine care is not standardised, and as such the decision to extubate is as usual clinical practice and not dictated by the protocol. We understand the limitations of this and on p13 lines 12-17 we acknowledge the limitation that the systems advice is not being compared to a well known standard of care so as to prevent comparing two interventional changes. We believe that this point is covered in that response, but are happy to consider it further.